# A Study on the Microscopic Properties of the Oil–Stone Interfacial Phase of a Reclaimed Asphalt Mixture Based on Molecular Dynamics Simulation

Yaoxi Cao [1], Yanhua Wang [2], He Li [3] and Wuxing Chen [4,*]

1   Jilin Communications Polytechnic, Changchun 130026, China; cyxi@jljy.edu.cn
2   Jinan City Planning and Design Institute, Jinan 250000, China; wyh-vivid@foxmail.com
3   Guidance and Service Center for Student Employment and Entrepreneurship, Jilin University, Changchun 130026, China; a326532558@jlu.edu.cn
4   School of Mines, China University of Mining and Technology, Xuzhou 221000, China
*   Correspondence: 6488@cumt.edu.cn

**Abstract:** In recent years, there has been a growing body of research focused on aged asphalt and recycled asphalt. Nevertheless, despite diligent endeavors, the precise micro-interaction mechanism occurring at the interface of weathered asphalt, reclaimed asphalt, and aggregates still eludes our understanding. This study leveraged molecular dynamics simulation technology to scrutinize the inherent behavior of aged asphalt and recycled asphalt at the micro-scale, elucidating the intricate interaction mechanism occurring at the interface of recycled asphalt, aged asphalt, and aggregates. The diffusion and adhesion properties of three distinct asphalt–aggregate interfaces were meticulously compared and comprehensively analyzed using advanced molecular dynamics simulation techniques. The findings revealed a substantial decline in the performance of aged asphalt, while the diffusion and adhesion properties of recycled asphalt were observed to be largely restored, reaching a remarkable level of approximately 85% in comparison to aged asphalt. The high-temperature performance, low-temperature performance, and water stability characteristics of both aged the asphalt mixture and recycled asphalt mixture were thoroughly investigated through rigorous laboratory testing of the asphalt mixtures. Based on the experimental findings, it was observed that the overall performance of the aged asphalt mixture exhibited a reduction of approximately 40%. However, through the process of regeneration, the overall performance of the asphalt mixture could be restored to approximately 90% of its non-aged counterpart.

**Keywords:** aged asphalt; recycled asphalt; molecular dynamics simulation; interface interaction analysis; laboratory test

## 1. Introduction

During the service process of asphalt, under the coupling effect of factors such as light, oxygen, water, and load, the lightweight components gradually transform into asphaltene, leading to asphalt aging and hardening, which lead to performance degradation [1–4]. Asphalt aging will lead to a series of problems such as road cracking, rutting, and potholes [5,6]. Asphalt aging seriously affects the service performance of pavement [7–10]. Therefore, how to restore aged asphalt to its initial state has become an important topic for scholars.

The aging problem of asphalt mixture has been studied by many scholars. Geng et al. [11] conducted pressure aging tests in a 0.2 MPa water vapor and 2.1 MPa oxygen environment, and the experimental results showed that water vapor can lead to the deterioration of the high-temperature stability and low-temperature crack resistance of asphalt. Vargas et al. [12] studied the test results of matrix asphalt after thermal oxygen aging in a reactor and found that the viscosity and particle size distribution of asphaltene increased with the increase in

thermal oxygen aging temperature. Zhang et al. [13] studied the rheological properties of different types of SBS-modified asphalt and found that the effect of aging on the properties of SBS-modified asphalt was related to the asphalt phase. Aging can reduce the phase angle and increase the complex modulus. Rodrigo et al. [14] simulated long-term aging to study the effect of aging on asphalt and found that the softening point and dynamic viscosity increased, while the penetration decreased. Soenen et al. [15] used laboratory simulation aging to study asphalt aging and found that during the aging process, the softening point increased by 13 °C and the penetration rate decreased by 60% at 25 °C. Liu et al. [16] conducted rheological and low-temperature performance tests on 70# base asphalt that was not aged and aged at 46 °C for 300 min, respectively. The results showed that this aging could increase the rutting factor by 4.5 times and improve the rutting resistance, but weakened the low-temperature performance.

Scholars have achieved extensive research results on the regeneration of aged asphalt. Zhang et al. [17] studied the effect of regenerants on the properties of aged asphalt and found that regenerants could restore the rheological properties of aged asphalt, but it was difficult to restore them to the original level, and different regenerants had varying degrees of impact on the crack resistance and adhesion of asphalt. Zaumanis et al. [18] selected six regenerants to regenerate RAP asphalt and found that compared to the original asphalt mixture, it exhibited good resistance to rutting and low-temperature cracking. Zhou et al. [19] studied the rutting resistance and fatigue resistance of SBS-modified recycled asphalt slurry and found that the recycling agent could weaken the rutting resistance of SBS-modified recycled asphalt and improve its fatigue life. Cong and Qiu et al. [20,21] found through experiments that for an SBS-modified asphalt mixture, the fatigue performance and low-temperature performance of the material improve after regeneration. Cong et al. [22] found through chromatography and other methods that regenerants can reduce the proportion of oxygen-containing groups in asphalt. Kuang et al. [23] believed that the addition of regenerant would cause a chemical reaction with aged asphalt. They believed that the lone pair in the amino group would a cause chemical reaction with functional groups in aged asphalt, such as aldehyde, carbonyl, etc., to form heavy components. Stimili et al. [24] used infrared spectroscopy to analyze the effect of the addition of waste SBS asphalt on the SBS content in different types of modified asphalt and found that the changes in SBS content varied among the different types. Zaumanis et al. [25] conducted an evaluation of the influence of six distinct recovery agents on the performance of recycled adhesives and blends. The findings indicated that vegetable oil showcased the most commendable performance, as its fatigue life nearly mirrored that of the original binder. In a study conducted by Im et al. [26], the impact of three regenerants on the mechanical properties of a recycled asphalt mixture, comprising 19% RAP (reclaimed asphalt pavement), was investigated. The results revealed that the inclusion of regenerants led to enhancements in the mixture's resistance to cracking and deformation while reducing its susceptibility to water damage. Chen et al. [27] found through research that commercial regenerants had better elastic recovery ability and deformation resistance than regenerants containing waste cooking oil. A. Ābele et al. [28] studied the regeneration of aged asphalt from biodiesel (F), the oxidation product of rapeseed oil, and showed that RTFOT modified by F-activated polymer had better low-temperature performance and fatigue resistance.

In recent years, laboratory research on aged asphalt has gradually increased, but the application of the regeneration of aged asphalt in actual road engineering is still limited. The main reason is that the micro-mechanical performance mechanism of the interface between recycled asphalt and mineral aggregates is still unclear. This study mainly used molecular dynamics simulation software to simulate and analyze the interaction mechanism and micro-mechanical properties of the interface between recycled asphalt and mineral aggregate, revealing the molecular-level interface behavior of the interface between recycled asphalt and mineral aggregate. This is the foundation for the application of recycled asphalt in road engineering.

## 2. Materials and Methods

### 2.1. Asphalt

The aged asphalt used in this study was extracted from an asphalt pavement in Northeast China for more than 3 years, and a basic performance test was carried out. The regeneration agent used in this study was 110# asphalt, which is rich in lightweight components. The basic performance indicators are shown in Table 1. The recycled asphalt in this article was obtained by mixing 110# asphalt with aged asphalt in a 1:1 ratio.

**Table 1.** Basic performance indicators of asphalt.

| Basic Indicator | 25 °C Penetration (0.1 mm) | 25 °C Ductility (cm) | Softening Point (°C) | Viscosity (pa·s) |
|---|---|---|---|---|
| Not aged | 62.7 | 80.2 | 68.5 | 1530 |
| Aged | 28.7 | 31.8 | 67.9 | 1920 |
| 110# | 109.2 | >100 | 43.1 | - |
| Test procedure | GB/T0606-2011 | GB/T0605-2011 | GB/T0606-2011 | GB/T0603-2011 |

### 2.2. Research Methods

This study used a combination of molecular dynamics simulation and experimental methods to study the regeneration mechanism of aged asphalt. A four-component test was carried out for aged asphalt, and its four-component composition was analyzed as the basis for molecular modelling. The four-component composition of aged asphalt is shown in Table 2.

**Table 2.** Four components of asphalt.

| Components | Saturated Phenol (%) | Aromatic Phenol (%) | Resin (%) | Asphaltene (%) |
|---|---|---|---|---|
| Not aged | 16.12 | 39.11 | 28.68 | 16.09 |
| Aged | 13.01 | 30.24 | 37.59 | 19.16 |
| 110# | 18.64 | 53.69 | 15.30 | 12.37 |

### 2.3. Aggregate

The aggregate selected in this study was alkaline stone, which came from Jiutai Stone Factory. The grading type selected in the experiment part of this study was AC-16, which is common in road surfaces, and the grading is shown in Figure 1. The oil/stone ratio is the best oil/stone ratio according to the calculation.

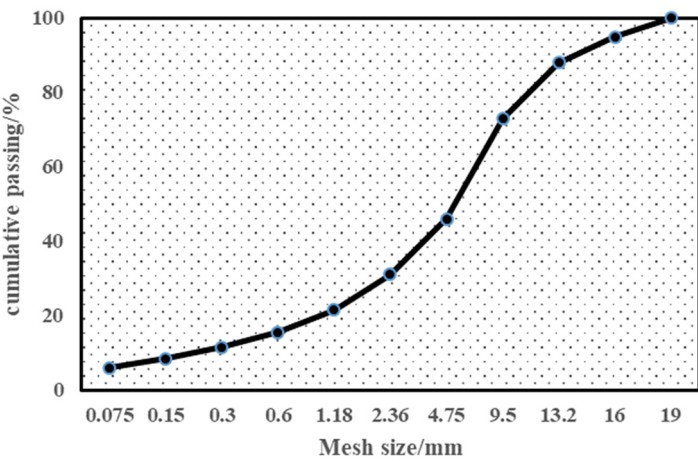

**Figure 1.** AC-16 particle size distribution.

### 3. Establishment and Rationality Verification of the Asphalt Molecular Model

We established molecular models of aged asphalt and recycled asphalt based on the results of asphalt four-component experiments and verified their rationality.

#### 3.1. Establishment of Asphalt Molecular Model

Asphalt is a complex mixture composed of hundreds of hydrocarbons [29]. The 12-component molecular model used in this article was developed by Li and Greenfield [30–33] through a series of studies and summaries, as shown in Figure 2.

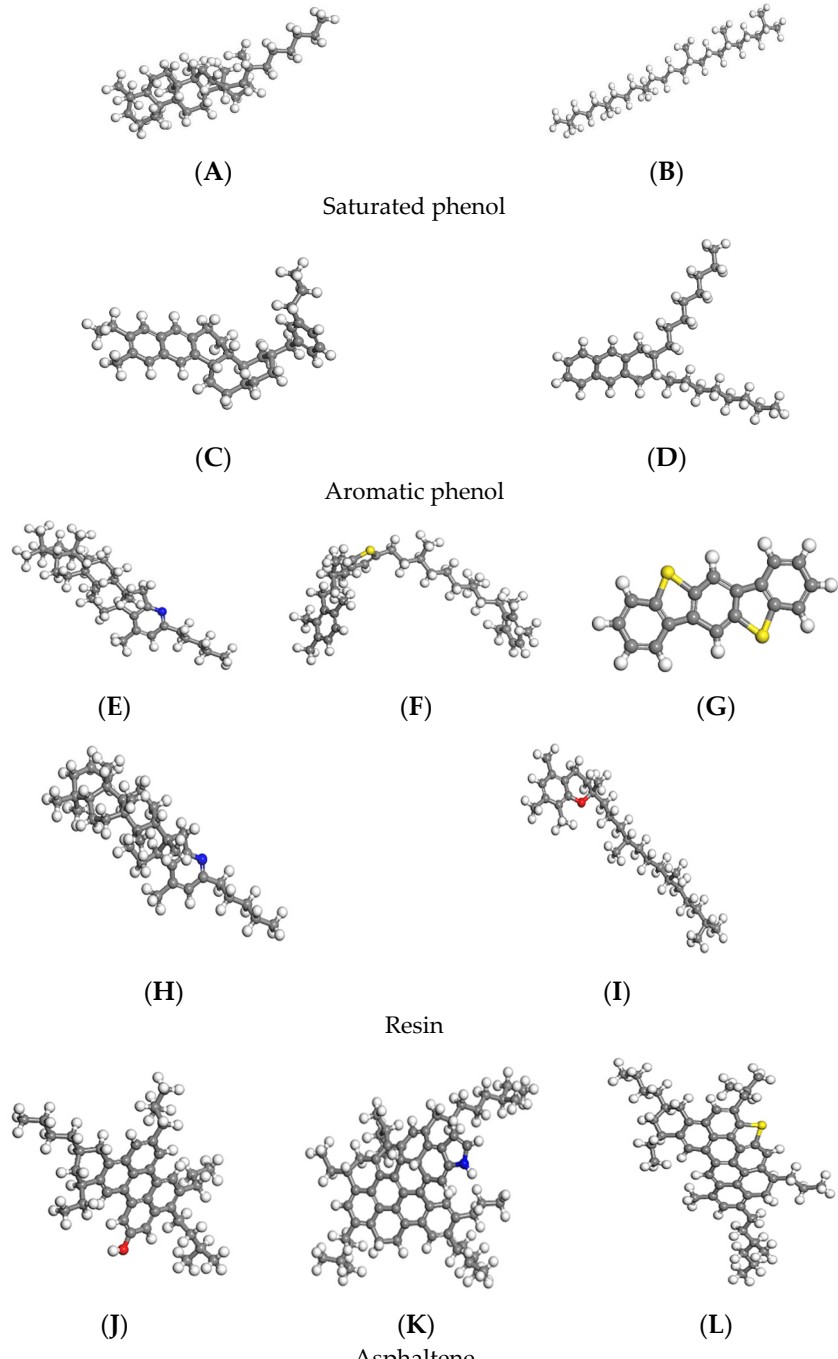

**Figure 2.** Asphalt component model.

This study aimed to establish three molecular models: non-aged asphalt, aged asphalt, and regenerated asphalt. The proportions of each component in each molecular model are shown in Table 3. We established the required asphalt molecular model based on the molecular models of each component of asphalt, as shown in Figure 3.

**Table 3.** Molecular composition of asphalt.

| Model Number | 110# | Not Aged | Aged | Regeneration |
|---|---|---|---|---|
| Saturate A | 11 | 11 | 9 | 10 |
| Saturate B | 10 | 10 | 8 | 9 |
| Aromatic C | 29 | 26 | 17 | 23 |
| Aromatic D | 34 | 28 | 19 | 26 |
| Resin E | 3 | 8 | 8 | 11 |
| Resin F | 3 | 6 | 6 | 4 |
| Resin G | 3 | 11 | 11 | 7 |
| Resin H | 3 | 8 | 8 | 6 |
| Resin I | 4 | 8 | 8 | 6 |
| Asphaltene J | 4 | 5 | 5 | 4 |
| Asphaltene K | 2 | 4 | 4 | 3 |
| Asphaltene L | 3 | 4 | 5 | 4 |

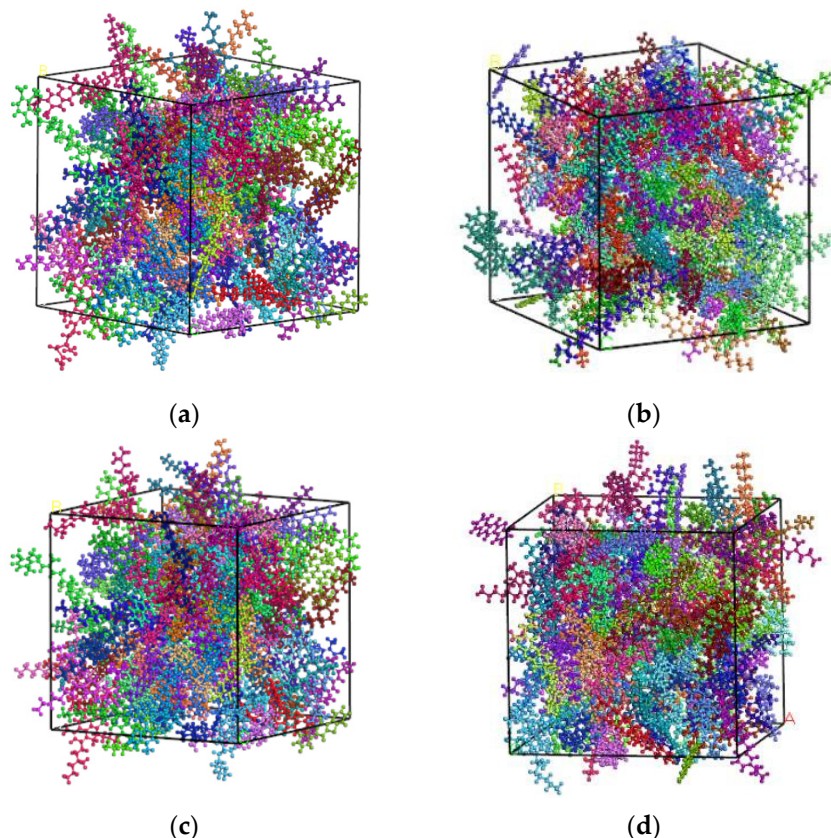

(**a**)　　　　　　　　　　　　　　　(**b**)

(**c**)　　　　　　　　　　　　　　　(**d**)

**Figure 3.** (**a**) 110# asphalt; (**b**) non-aged asphalt; (**c**) aged asphalt; (**d**) recycled asphalt.

### 3.2. Establishment of Mineral Crystal Models

Limestone and basalt are the most commonly used aggregates in asphalt road engineering, and their main component is $SiO_2$. Therefore, in this article, $SiO_2$ crystals were used to represent the surface of mineral crystals [34]. The crystal bond length and bond angle of $SiO_2$ are a = 4.913 Å, b = 4.913 Å, c = 5.405 Å, $\alpha = 90°$, $\beta = 90°$, and $\gamma = 120°$. The established $SiO_2$ crystal cell and crystal model are shown in Figure 4.

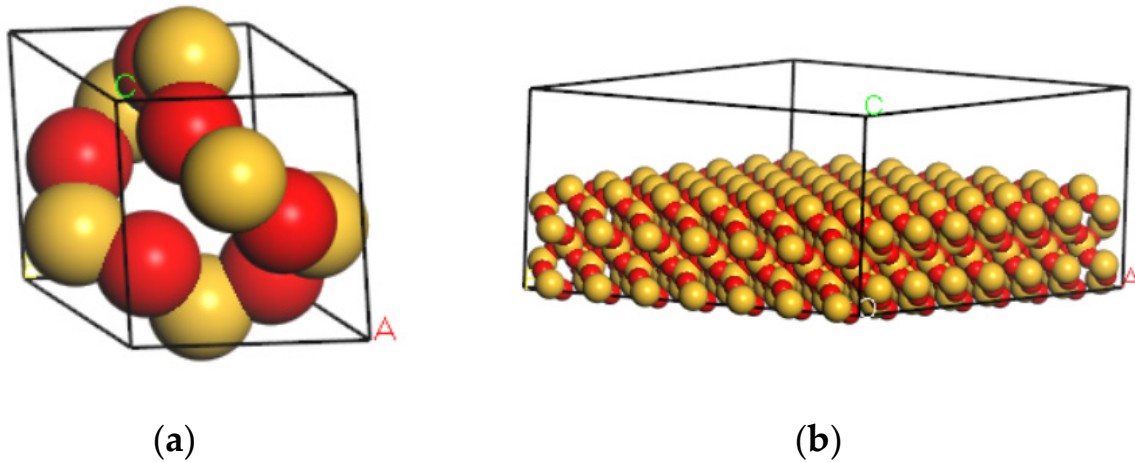

**(a)** **(b)**

**Figure 4.** (**a**) SiO$_2$ crystal cell; (**b**) SiO$_2$ crystal model.

### 3.3. Verification of the Rationality of the Asphalt Model

3.3.1. Density

The rationality verification of the asphalt molecular model included two indicators, with density being one of them. The density of asphalt molecules is an important physical indicator, and there must be a difference between the model density and the actual asphalt density. If the difference is within an acceptable range, the model is considered reasonable. The actual density and model density of the four types of asphalt in this article are shown in Table 4.

**Table 4.** Asphalt model density and actual density at 0 °C.

| Asphalt | Model Density Value (g/cm$^3$) | Actual Density Value (g/cm$^3$) | Density Ratio (%) |
|---|---|---|---|
| 110# | 0.963 | 0.987 | 97.6 |
| Not aged | 1.014 | 1.042 | 97.3 |
| Aged | 1.025 | 1.055 | 97.2 |
| Regeneration | 1.011 | 1.031 | 98.1 |

According to the data in Table 4, it can be seen that the model density was above 97% of the actual density, indicating that the model could represent the actual asphalt, and the established model was reasonable.

3.3.2. Solubility Parameters

The solubility parameter is an indicator that reflects the compatibility between various components. The smaller the difference in solubility parameters between each component, the better the solubility, and a blend system can be formed. Generally, the difference in solubility parameters between each component should be less than 4.1 (J/cm$^3$)$^{1/2}$, and each component can form a blend system. The square root of the cohesive energy density is the solubility parameter. The solubility parameters of various asphalt components are shown in Table 5.

From the data in Table 5, it can be seen that the intragroup difference in solubility parameters of each asphalt component was less than 4.1 (J/cm$^3$)$^{1/2}$, indicating that various components could be mixed into a system, indicating that the established asphalt model was reasonable. The density and solubility parameters were verified to be within a reasonable range, indicating that the established asphalt model was acceptable.

**Table 5.** Solubility parameters.

| Asphalt | Components | Cohesive Energy Density/(J/m$^3$) | Solubility Parameter/(J/cm$^3$)$^{1/2}$ | Electrostatic Solubility Parameter/(J/cm$^3$)$^{1/2}$ | Van der Waals Solubility Parameter/(J/cm$^3$)$^{1/2}$ |
|---|---|---|---|---|---|
| 110# | Saturate | $2.582 \times 10^8$ | 17.623 | 2.220 | 17.515 |
| | Aromatic | $2.937 \times 10^8$ | 19.345 | 1.389 | 18.831 |
| | Resin | $3.023 \times 10^8$ | 16.848 | 3.509 | 16.528 |
| | Asphaltene | $3.345 \times 10^8$ | 18.213 | 1.072 | 17.778 |
| | Intragroup difference | $7.630 \times 10^7$ | 2.497 | 2.437 | 2.303 |
| Not aged | Saturate | $2.601 \times 10^8$ | 16.945 | 1.760 | 16.058 |
| | Aromatic | $2.718 \times 10^8$ | 18.574 | 0.855 | 18.235 |
| | Resin | $3.148 \times 10^8$ | 17.390 | 2.692 | 17.010 |
| | Asphaltene | $3.923 \times 10^8$ | 19.711 | 3.301 | 18.191 |
| | Intragroup difference | $1.312 \times 10^8$ | 2.766 | 2.466 | 2.177 |
| Aged | Saturate | $2.876 \times 10^8$ | 17.562 | 1.841 | 17.366 |
| | Aromatic | $3.125 \times 10^8$ | 16.368 | 3.681 | 16.031 |
| | Resin | $3.487 \times 10^8$ | 18.048 | 3.260 | 17.901 |
| | Asphaltene | $4.022 \times 10^8$ | 18.509 | 1.648 | 17.979 |
| | Intragroup difference | $1.146 \times 10^8$ | 2.141 | 2.033 | 1.948 |
| Regeneration | Saturate | $2.520 \times 10^8$ | 16.885 | 1.813 | 16.123 |
| | Aromatic | $2.619 \times 10^8$ | 18.781 | 0.912 | 18.435 |
| | Resin | $3.098 \times 10^8$ | 17.098 | 2.776 | 17.005 |
| | Asphaltene | $3.877 \times 10^8$ | 19.821 | 3.419 | 18.226 |
| | Intragroup difference | $1.357 \times 10^8$ | 2.936 | 2.507 | 2.312 |

## 4. Molecular Dynamics Simulation and Experimental Verification of the Interface between Asphalt and Mineral Crystals

### 4.1. Establishment of an Interface Model between Asphalt and Aggregate

We assembled the three established asphalt models and mineral crystal surface models into interface models. The model was divided into three layers, including a mineral crystal layer, asphalt layer, and vacuum layer, with a total of 30 Å and 10 Å for each layer. The interface models between three types of asphalt and mineral crystal surfaces established are shown in Figure 5.

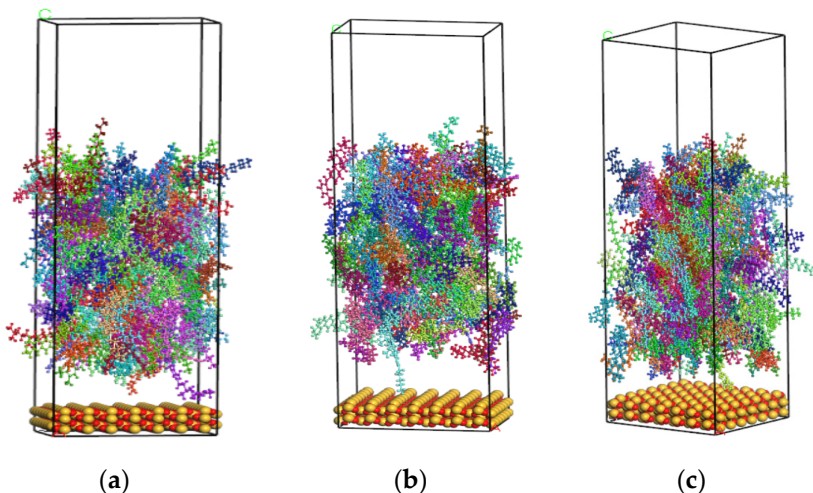

(**a**)　　　　(**b**)　　　　(**c**)

**Figure 5.** (**a**) Non-aged asphalt–aggregate; (**b**) aged asphalt–aggregate; (**c**) recycled asphalt–aggregate.

### 4.2. Dynamic Simulation and Interface Energy Analysis of Asphalt–Aggregate Interface

The established interface model was subjected to structural optimization and annealing to eliminate energy singularities. The selected force field was COMPASS, and the ensemble was NVT and NPT. We performed a dynamic simulation on the optimized interface model at temperatures of −25 °C, 0 °C, and 25 °C. Figure 6 shows the energy changes during the dynamic simulation process.

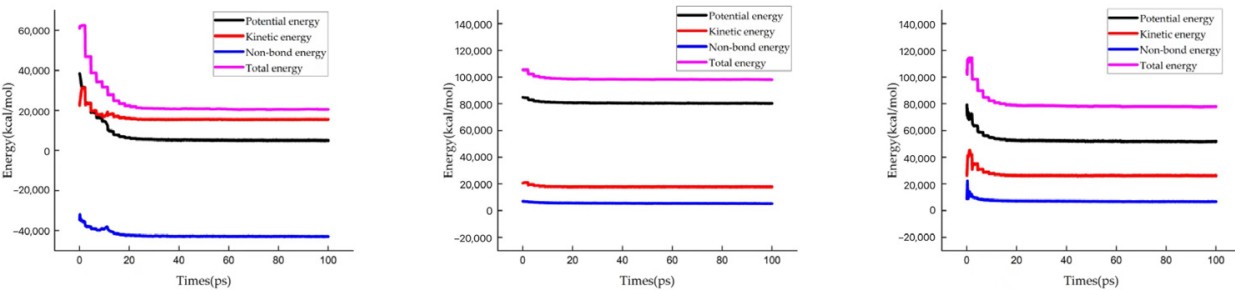

**Figure 6.** Dynamic simulation process energy changes.

The formula for calculating the interfacial energy between asphalt and aggregate is shown in Formula (1), where $E_{Interface}$ is the interfacial energy, $E_{Dynamics}$ is the system energy after the dynamic simulation, and $E_{Anneal}$ is the system energy after annealing. The simulation calculation results of the interface energy between three types of asphalt and aggregate are shown in Table 6, and the data are plotted in Figure 7.

$$E_{Interface} = E_{Dynamics} - E_{Anneal} \tag{1}$$

**Table 6.** Interface energy calculation results.

| Temperature (°C) | Interface | $E_{Dynamics}$ (kcal/mol) | $E_{Anneal}$ (kcal/mol) | $E_{Interface}$ (kcal/mol) |
|---|---|---|---|---|
| −25 °C | Non-aged asphalt–aggregate | 8971.98264 | 43,393.23788 | −34,421.25524 |
| | Aged asphalt–aggregate | 8299.21359 | 36,688.33748 | −28,389.12389 |
| | Recycled asphalt–aggregate | 8722.37459 | 41,592.66435 | −32,870.28976 |
| 0 °C | Non-aged asphalt–aggregate | 9701.25834 | 43,411.24584 | −33,709.98754 |
| | Aged asphalt–aggregate | 8123.25683 | 35,341.47055 | −27,218.21372 |
| | Recycled asphalt–aggregate | 8966.89427 | 39,844.33987 | −30,877.44566 |
| 25 °C | Non-aged asphalt–aggregate | 10,020.87298 | 43,336.85935 | −33,315.98637 |
| | Aged asphalt–aggregate | 7768.99356 | 34,401.2026 | −26,632.25673 |
| | Recycled asphalt–aggregate | 9879.54937 | 39,962.99473 | −30,083.44536 |

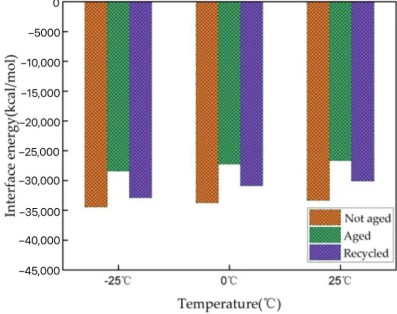

**Figure 7.** Interface energy.

We calculated the interface energy of different types of asphalt aggregate interface models through dynamic simulation. Subtracting the system energy before the simulation from the system energy the after dynamic simulation gave the energy required to form the interface, which was the interface energy. Negative numbers represent attraction. The larger the absolute value of the calculated data in the table, the greater the interface energy

and the better the adsorption strength. According to the data in Table 6 and Figure 5, it can be concluded that the interface energy between aged asphalt and aggregate was the lowest, while the interface energy between asphalt and aggregate was the highest. The interface energy between recycled asphalt and aggregate was located in the middle. This indicates that the proportion of each component changed after asphalt aging, causing it to become brittle and hard, resulting in a decrease in adhesion performance. At −25 °C, the interfacial energy between the aged asphalt and aggregate decreased by 17.5% compared to that of the non-aged asphalt aggregate. At 0 °C, the interfacial energy between the aged asphalt and aggregate decreased by 19.3% compared to that of the non-aged asphalt aggregate. At 0 °C, the interfacial energy between the aged asphalt and aggregate decreased by 19.3% compared to that of the non-aged asphalt aggregate, and the interfacial energy between the aged asphalt and aggregate decreased by 20.1%. For the recycled asphalt, at a temperature of −25 °C, the interface energy between the recycled asphalt and aggregate could reach 95.5% compared to that of the non-aged asphalt aggregate. At 0 °C, the interface energy between the recycled asphalt and aggregate could reach 91.6% compared to that of the non-aged asphalt aggregate. At 25 °C, the interface energy between the recycled asphalt and aggregate could reach 90.3% compared to that of the non-aged asphalt aggregate. For the recycled asphalt, the interfacial energy between the recycled asphalt and aggregate could reach 95.5% at −25 °C compared to that of the non-aged asphalt aggregate. At a temperature of 0 °C, compared to the interface energy of the non-aged asphalt aggregate, the interface energy between recycled asphalt and aggregate can reach 91.6%. At a temperature of 25 °C, compared to the interface energy of the non-aged asphalt aggregate, the interface energy between the recycled asphalt and aggregate could reach 90.3%. The 110# asphalt supplemented the lightweight components that were missing from the aged asphalt, restoring its vitality and restoring its interfacial adhesion performance.

*4.3. Analysis of the Diffusion Performance of Asphalt on the Surface of Aggregates*

Liquid and gas molecules do not stay in fixed positions but continue to move. Mean square displacement (*MSD*) refers to the average square of the atomic displacement in a system at any time t. By calculating the mean square displacement, the diffusion law of asphalt on the surface of aggregates can be analyzed. The calculation method for mean square displacement is shown in Formula (2).

$$MSD(t) = \frac{1}{N} \sum_{i=1}^{N} [r_i(t) - r_i(0)]^2 \tag{2}$$

where $r_i(t)$ is the particle displacement at time $t$, $r_i(0)$ is the particle displacement at the initial time, and $N$ is the particle number. The larger the slope of the *MSD* curve, the greater the particle mobility, and the better the diffusion performance. The *MSD* of three types of asphalt on the aggregate surface at −25 °C, 0 °C, and 25 °C is shown in Figure 8.

According to the data in Figure 7, it can be seen that as the temperature increased, the diffusion rates of the three types of asphalt on the surface of the aggregate increased, which was due to the increase in temperature increasing the activity of asphalt molecules. Regardless of the temperature, the diffusion performance of the aged asphalt on the surface of the aggregate was the worst, while the diffusion performance of the recycled asphalt on the surface of the aggregate after activation with a regenerant was similar to that of the non-aged asphalt. This is because recycled asphalt is supplemented with lightweight components, which mainly serve as wetting agents at the interface between asphalt and aggregates, thus restoring the diffusion performance of recycled asphalt on the surface of aggregates.

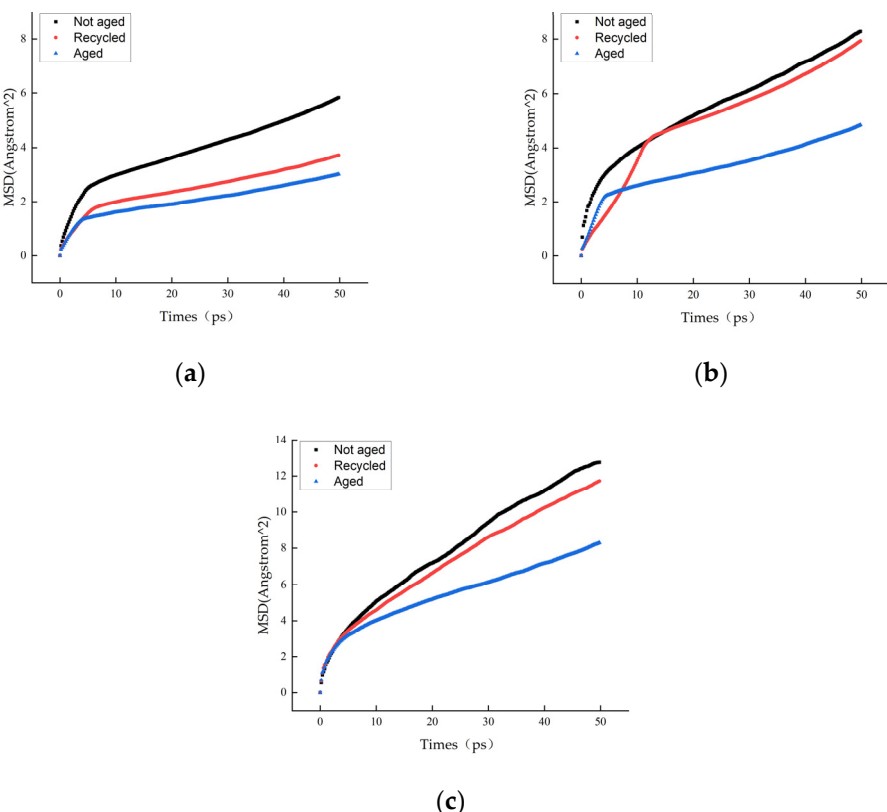

**Figure 8.** (**a**) −25 °C *MSD*; (**b**) 0 °C *MSD*; (**c**) 25 °C *MSD*.

*4.4. High-Temperature Performance Analysis of Recycled Asphalt Mixture*

4.4.1. Marshall Stability

A Marshall stability test was used in this study to compare and analyze the high-temperature stability performance of the aged asphalt mixture, non-aged asphalt mixture, and recycled asphalt mixture. The experimental process is shown in Figure 9, and the experimental results are shown in Figure 10.

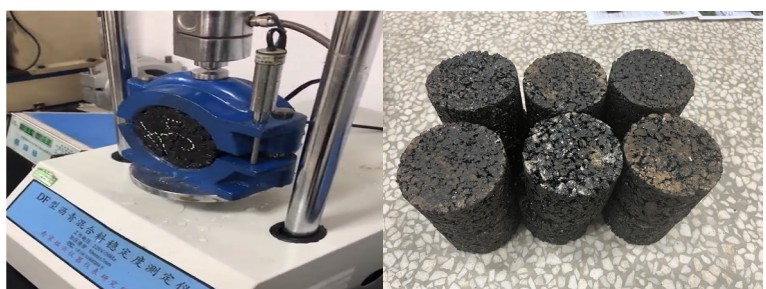

**Figure 9.** Marshall stability test.

According to the Marshall stability experimental data in Figure 10, it can be seen that the Marshall stability of the asphalt mixture decreased significantly after aging, and the Marshall stability of the recycled asphalt mixture was greatly improved. After aging, the Marshall stability of the asphalt mixture decreased by 31.2%, while after regeneration, the Marshall stability of the asphalt mixture recovered to 91.5%. This shows that the addition of 110# asphalt can effectively restore the high-temperature performance of an aged asphalt mixture, mainly because the addition of a regenerator can restore the composition ratio of the aged asphalt and restore the activity of the aged asphalt.

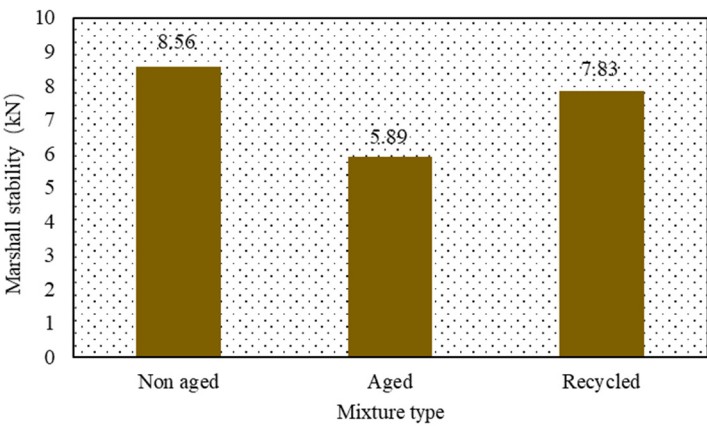

**Figure 10.** Marshall stability.

4.4.2. Rutting Test

Dynamic stability is an important index to test the high-temperature performance of asphalt mixtures. In this study, the difference in high-temperature performance of three asphalt mixtures was studied through a rutting test. The experimental process is shown in Figure 11, and the test results are shown in Figure 12.

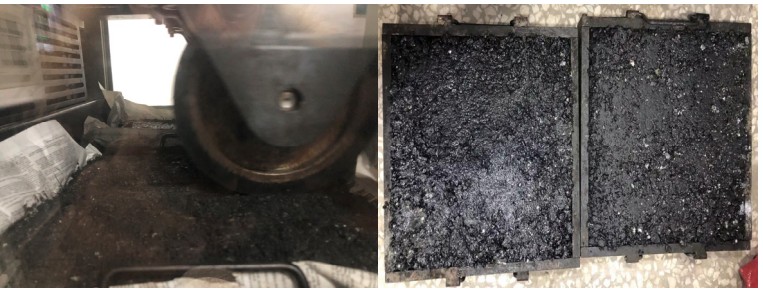

**Figure 11.** Rutting test.

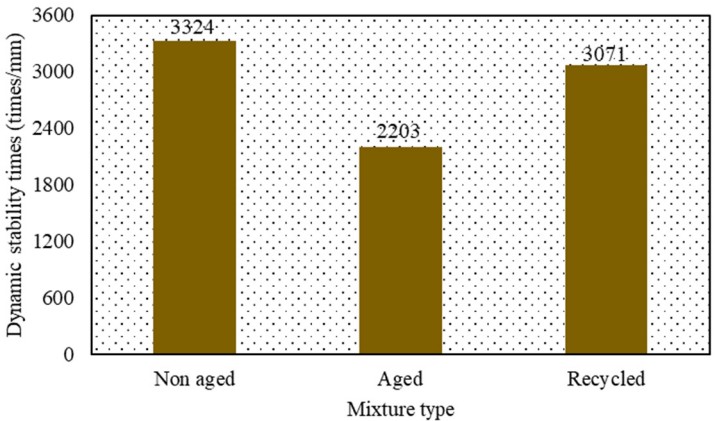

**Figure 12.** Dynamic stability times.

As can be seen from the rutting experiment data in Figure 12, the dynamic stability frequency of the asphalt mixture after aging decreased significantly, while the dynamic stability frequency of the recycled asphalt mixture was greatly improved compared with that of the aged asphalt mixture. After aging, the dynamic stability of the asphalt mixture decreased by 33.7%, and after regeneration, the dynamic stability of the asphalt mixture recovered to 92.4% of the non-aged mixture. The rutting test results fully show that the rutting resistance of aged asphalt was significantly improved after regeneration.

### 4.5. Low-Temperature Performance Analysis of the Recycled Asphalt Mixture

### 4.5.1. Low-Temperature Splitting Test

In this study, a low-temperature splitting test was used to study the difference in low-temperature cracking resistance of three asphalt mixtures. The experimental process is shown in Figure 13, and the experimental results are shown in Figure 14.

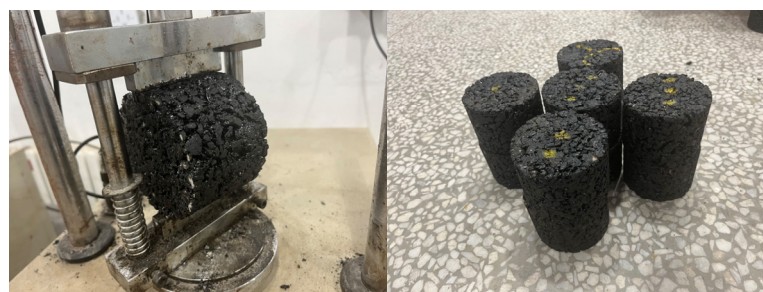

**Figure 13.** Low-temperature splitting test.

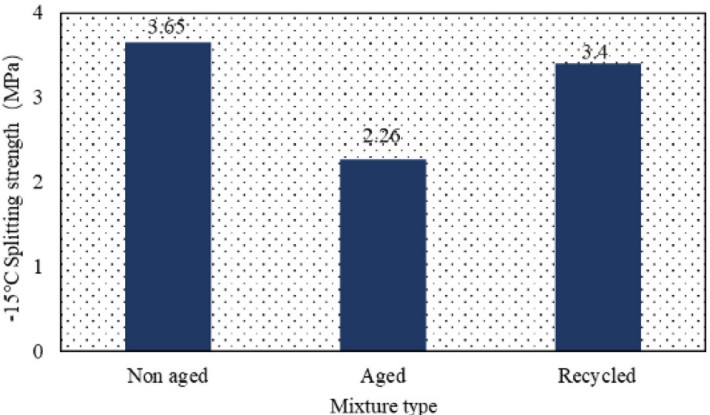

**Figure 14.** Splitting strength.

As can be seen from the data of the low-temperature splitting test in Figure 14, the low-temperature splitting strength of the asphalt mixture after aging was significantly reduced, while the low-temperature splitting strength of the recycled asphalt mixture was significantly increased compared with that of the aged asphalt mixture. After aging, the low-temperature splitting strength of the asphalt mixture decreased by 38.1%, and after regeneration, the low-temperature splitting strength of the asphalt mixture recovered to 93.2% of that without aging. The results of the low-temperature splitting test fully show that the low-temperature cracking resistance of aged asphalt was significantly improved after regeneration.

### 4.5.2. Low-Temperature Trabecular Bending Failure Test

At low temperatures, an asphalt mixture can be regarded as an elastic material, and its destruction process can be regarded as a process of energy dissipation. When external forces exert a certain amount of work on it, the asphalt mixture stores the energy. As the accumulation of stored energy reaches a certain limit, the asphalt mixture will crack, and the stored energy will be converted into surface energy dissipation. The higher the storage capacity of elastic strain energy of the asphalt mixture, the better its low-temperature cracking resistance. The trabecular size used in this study was 250 mm × 30 mm × 35 mm and the experimental temperature was −10 °C. The low-temperature trabecular bending experimental process of the three asphalt mixtures is shown in Figure 15, and the experimental results are shown in Figure 16.

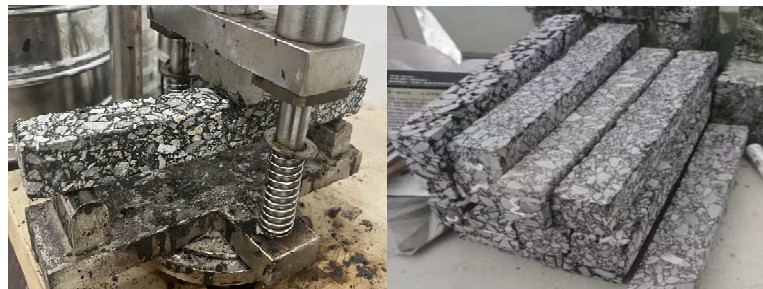

**Figure 15.** Low-temperature trabecular bending failure test.

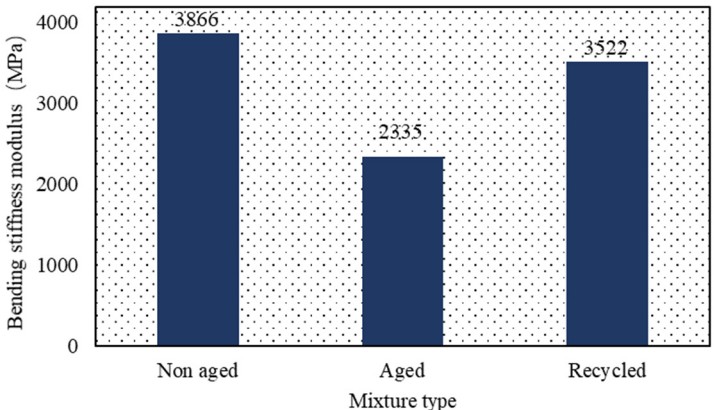

**Figure 16.** Bending stiffness modulus.

According to the experimental data in Figure 16, it can be seen that the ability of the asphalt mixture to store deformation at a low temperature decreased greatly after aging, which made it more prone to crack. After aging, the bending stiffness modulus of the asphalt mixture was reduced by 39.6%, while after regeneration, the bending stiffness modulus of the asphalt mixture was restored to 91.1% of that before aging. The test results fully show that the storage deformation ability of the aged asphalt mixture was restored after regeneration at low temperature, and the low-temperature crack resistance was improved.

*4.6. Moisture Sensitivity Analysis of Recycled Asphalt Mixture*

4.6.1. Freeze–Thaw Cycle Splitting Test

In this study, the samples were freeze–thawed for 14 days, and then a splitting test was carried out to compare the difference in water stability of the three asphalt mixtures. The period of freezing for 12 h and melting for 12 h was a cycle. The experimental process is shown in Figure 17, and the experimental results are shown in Figure 18.

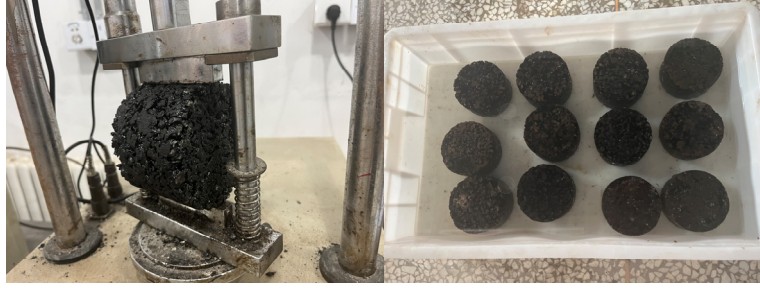

**Figure 17.** Freeze–thaw cycle splitting test.

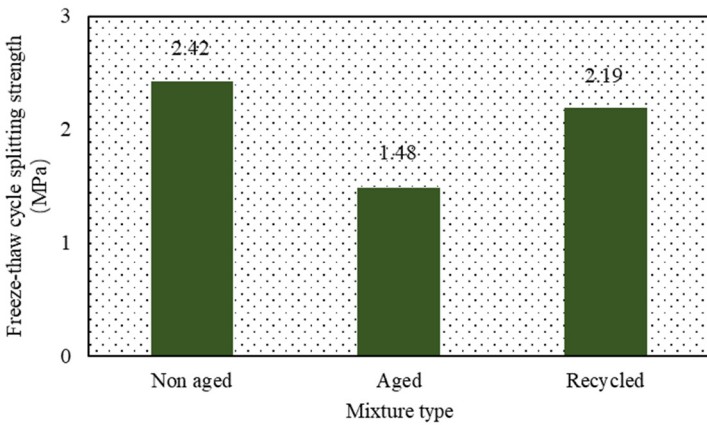

**Figure 18.** Freeze–thaw cycle splitting strength.

According to the test data in Figure 18, it can be seen that the splitting strength of the aged asphalt mixture decreased significantly after 14 d of the freeze–thaw cycle. The freeze–thaw cyclic splitting strength of the aged asphalt mixture was 38.8% lower than that of the non-aged asphalt mixture, while the freeze–thaw cyclic splitting strength of the regenerated asphalt mixture could reach 90.5% of that of the non-aged asphalt mixture. The results of the freeze–thaw cycle splitting test fully show that the water stability of the aged asphalt mixture was significantly restored after regeneration.

4.6.2. Dynamic Water Erosion Splitting Test

In the rainy season, under the repeated action of vehicle load, rain will enter asphalt mixtures and repeatedly wash their internal structure. Frequent dynamic water washing and pumping in the pores lead to the physical separation of the aggregate and asphalt binder film, resulting in the loss of strength and stiffness of the asphalt mixture. In this study, the samples were washed in a dynamic water washing machine, and then a splitting test was carried out to compare the water stability difference of the three asphalt mixtures. The dynamic water washing included 3600 times per cycle. The experimental process is shown in Figure 19, and the experimental results are shown in Figure 20.

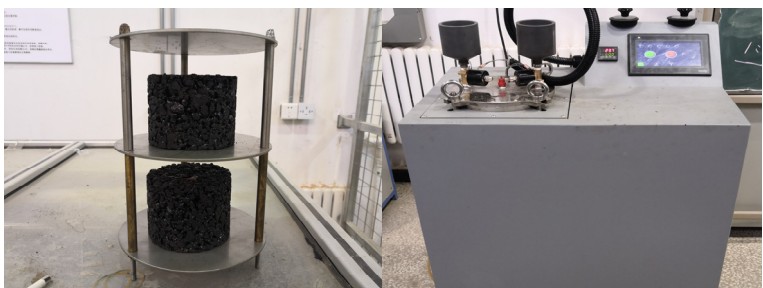

**Figure 19.** Dynamic water erosion splitting test.

It can be seen from the test data in Figure 20 that the splitting strength of the aged asphalt mixture decreased significantly after a cycle of dynamic water scouring. The dynamic water erosion splitting strength of the aged asphalt mixture was 43.3% lower than that of the non-aged asphalt mixture, while the dynamic water erosion splitting strength of the reclaimed asphalt mixture could reach 89.2% of that of the non-aged asphalt mixture. The results of the dynamic water erosion and splitting test fully show that the water stability of the aging asphalt mixture was obviously restored after regeneration.

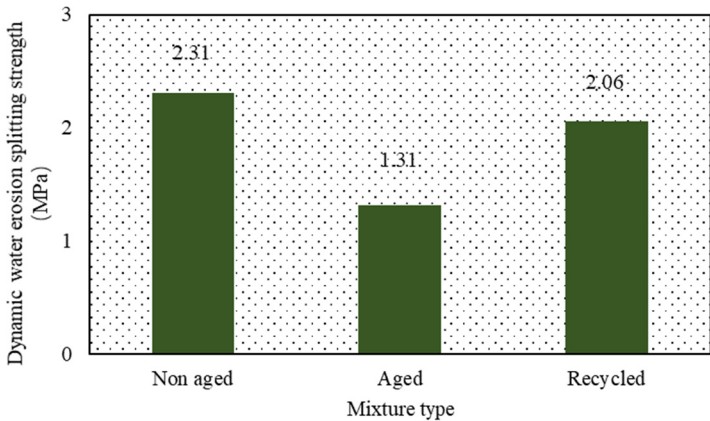

**Figure 20.** Dynamic water erosion splitting strength.

## 5. Conclusions

This study conducted in-depth research on the interface interaction mechanism between recycled asphalt and aggregates using molecular dynamics simulation and experimental methods. It has theoretical guidance and practical significance for asphalt mixture regeneration. The main conclusions are as follows:

(1) A molecular model of the interface between aged asphalt and aggregate was established, and the reason for poor interface performance between aged asphalt and aggregate was analyzed through microscopic-scale analysis.

(2) Based on molecular dynamics simulation, the diffusion and adhesion properties of the aged asphalt and aggregate interface, recycled asphalt and aggregate interface, and non-aged asphalt and aggregate interface were compared and analyzed at the micro-scale. The results showed that the diffusion and adhesion properties of the recycled asphalt were restored, reaching over 85% of that of the non-aged asphalt.

(3) The high-temperature performance, low-temperature performance, and water stability performance of the asphalt mixture were studied through an asphalt mixture laboratory test. The experimental results showed that the performance of the asphalt mixture decreased after aging, but recovered effectively after regeneration. The experimental results of the asphalt mixtures and the simulation results of molecular dynamics were mutually confirmed. The high-temperature performance, low-temperature performance, and water stability performance of the reclaimed asphalt mixture could be restored to more than 90% of that before aging.

**Author Contributions:** Conceptualization, W.C.; software, Y.C.; formal analysis, Y.C.; investigation, H.L.; data curation, Y.C. and Y.W.; writing—original draft preparation, Y.C.; writing—review and editing, W.C.; project administration, W.C. All authors have read and agreed to the published version of the manuscript.

**Funding:** This research was funded by the National Nature Science Foundation of China (NSFC) (Grant No. 51178204).

**Institutional Review Board Statement:** Not applicable.

**Informed Consent Statement:** Not applicable.

**Data Availability Statement:** Not applicable.

**Acknowledgments:** This research was funded by the National Natural Science Foundation of China (NSFC) (Grant No. 51178204). This financial support is gratefully acknowledged.

**Conflicts of Interest:** The authors declare no conflict of interest.

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
