# Peer review of "A Study on the Microscopic Properties of the Oil–Stone Interfacial Phase of a Reclaimed Asphalt Mixture Based on Molecular Dynamics Simulation"

_coatings, doi:10.3390/coatings13101717_

Round 1
Reviewer 1 Report
The manuscript is acceptable
Moderate editing of English language required
Reviewer 2 Report
It is very positive that the research shows a connection between the composition of bitumen and its effect on the properties of asphalt concrete, because these results can contribute to the development of a more effective recycling technology. A few notes:
1. In line 35, it should be specified what "mud boiling" is, because such a term is not traditionally used to describe asphalt damage
2. In line 40, what temperature was used in the PAV test
3. Preferably 2.1. in the chapter to explain the place where asphalt concrete was used for 3 years, in order to have a connection with a place and the climate characteristic of that place.
4. Line 215 should most likely read "not aged" instead of "aged".
5. Section 4.5.2 could describe the sample dimensions, loading conditions and temperature in more detail
6. The conclusions should include more numerical values ​​so that it is clear what effect (in numbers, %) can be achieved on the bitumen aggregate interface after rejuvenation
The article is relevant, innovative and of great practical importance (regenerator/rejuvenator diffusion and the stability of the rejuvenated system, as well as the adhesions between rejuvenated bitumen and aggregate are still a problem that needs to be solved), but in order to emphasize innovation even more, it would be desirable to increase the number of reference articles that are not older than 5 years in relevant high-ranking journals, for example, referring to the DOI: 10.1007/s42947-020-0051-y
Reviewer 3 Report
This research has focused on the microscale behavior of aged and recycled asphalt. Molecular dynamics simulations revealed a decline in aged asphalt performance, while recycled asphalt showed improved properties, reaching 85% of its original state. Lab tests indicated a 40% reduction in aged asphalt mixture performance, but regeneration can restore it to about 90% of its original state, highlighting the significance of understanding these microscale processes for asphalt durability and sustainability in construction and infrastructure. This study is interesting but there is something to revise.
1. In Section 1 introduction, what are the scholars? it is not a conventional expression of that in the paper. please modify the sentences.
2. Section 2.3. it is better to show the figure of particle size distribution instead of table 3.
3. In Table 1, the font style of the unit for temp. is different. please unify it.
4. In Figure 1. Asphalt component model, please indicate and distinguish what the molecular is A or B.
5. In Table 4, The text alignment inside the table is not correct.
6. In Table 7. There are too many significant figures; I recommend reducing them. Also, the content within the interface tab looks complex.
7. please check the font style of the title in Figure 7
8. In Figure 9, KN -> kN
9. In Figure 11, what is the times/mm? for a minute, an hour?
10. In Figures 13, 15, 17, and 19, Mpa -> MPa
11. The conclusion is too short. what is the purpose of this research? what is the limitation of this research?
12. References are too old, please update the new ones.
I have some parts in the sentence that I don't understand. It looks like it needs some English corrections.
